# Comparative Analysis of Anterolateral and Posterior Approaches for Distal Humerus Shaft Fractures: A Multicenter Retrospective Study

**DOI:** 10.3390/jcm14092890

**Published:** 2025-04-22

**Authors:** Yong-Cheol Yoon, Hyoung-Keun Oh, Hyung-Suh Kim, Joon-Woo Kim

**Affiliations:** 1Orthopedic Trauma Division, Trauma Center, Gachon University College of Medicine, Namdong-gu, Incheon 21565, Republic of Korea; dryoonyc@gmail.com; 2Department of Orthopedic Surgery, Inje University Ilsan Paik Hospital, Ilsanseo-gu, Goyang-si 10380, Gyeonggi-do, Republic of Korea; 2103095@naver.com; 3Department of Orthopedic Surgery, School of Medicine, Kyungpook National University, Kyungpook National University Hospital, Jung-gu, Daegu 41944, Republic of Korea; joonwoo@knu.ac.kr

**Keywords:** distal humerus shaft fracture, anterolateral approach, posterior triceps-sparing approach, radial nerve, functional outcomes, bone union

## Abstract

**Background**: Distal humeral shaft fractures (DHSFs) pose surgical challenges due to the proximity to the elbow joint, limited bone stock, and the risk of radial nerve injury. This study compared clinical and radiographic outcomes of anterolateral and posterior triceps-sparing approaches to determine the most effective surgical strategy. **Methods**: This multicenter retrospective study included 75 patients who underwent surgery for a DHSF between 2015 and 2021, with a minimum one-year follow-up, a distal fragment ≥3 cm, and no preoperative radial nerve injury. Fifty patients underwent anterior plating via anterolateral approach, and twenty-five underwent posterior plating. Clinical and radiographic outcomes were evaluated. **Results:** Bone union was achieved in 74 patients (98.7%), with no significant difference between the groups (*p* = 0.21). The anterolateral approach resulted in a shorter operative time (116 ± 29.4 vs. 143 ± 31.4 min, *p* = 0.03). However, intraoperative blood loss (*p* = 0.36), Mayo Elbow Performance Score (*p* = 0.71), range of motion (*p* = 0.36), and complication rates (*p* = 0.21) were not significantly different. Two cases of transient radial nerve palsy occurred in the posterior group (*p* = 0.17), and four cases required implant removal due to discomfort (*p* = 0.18) in the anterolateral group. **Conclusions**: Both approaches effectively treat DHSFs with high union rates and comparable functional outcomes. However, the anterolateral approach significantly reduces operative time due to supine positioning, direct access, and avoiding radial nerve dissection. Posterior plating remains viable when stable anterior fixation is unachievable. Further studies should assess the long-term outcomes and factors influencing approach selection.

## 1. Introduction

A distal humeral shaft fracture (DHSF) is an extra-articular fracture located in the distal third of the humerus, accounting for approximately 2% of all fractures [1,2]. The humeral shaft extends from the distal region of the surgical neck to the proximal margin of the supracondylar ridge, with DHSF typically centered in the distal one-third of the humerus [3,4].

Treatment strategies for DHSF remain a topic of debate, as both conservative and surgical approaches have been proposed. Sarmiento et al. [5] reported that conservative management with functional bracing led to a high rate of bone union and satisfactory functional outcomes. However, Jawa et al. [6] highlighted the limitations of bracing, noting that surgical fixation provides better alignment, faster recovery, and reduces the risk of complications, including skin irritation and malalignment. Given the challenges in maintaining proper alignment in the transitional zone of the humerus, Schatzker et al. [7] suggested that rigid fixation is necessary to facilitate early mobilization and prevent joint stiffness.

Among surgical approaches, the posterior approach with posterior plating has been widely utilized. This approach allows for direct visualization and exploration of the radial nerve, adequate exposure of the fracture site, and stable fixation [8,9,10]. However, concerns have been raised regarding iatrogenic radial nerve injury, soft tissue irritation from metal implants, elbow range of motion (ROM) limitations, and difficulties with implant removal due to soft tissue adhesions [11].

Alternatively, the anterolateral approach with anterior plating has been introduced as a minimally invasive technique that avoids direct radial nerve exposure, minimizes surgical trauma, and reduces the risk of nerve damage [12]. However, this technique has some limitations. The anterior approach provides limited space for plating, which can compromise fixation stability, particularly when the distal fragment is small [12]. Moreover, concerns remain regarding the biomechanical strength of anterior plating compared with that of posterior plating [13].

Although both surgical approaches are commonly used in clinical practice, there is no clear consensus on the optimal approach for DHSF treatment. Although previous studies examined individual techniques, comparative studies evaluating their relative advantages and limitations remain limited.

This study aimed to directly compare the anterolateral and posterior approaches for DHSF by analyzing bone union rates, functional outcomes, and complications associated with each technique. By evaluating key surgical outcomes in a multicenter retrospective setting, we sought to establish evidence-based recommendations for the optimal surgical approach for DHSF.

## 2. Materials and Methods

### 2.1. Demographics

Among the patients diagnosed with a DHSF who underwent surgical treatment between March 2015 and December 2021 in three level I trauma centers, 75 (47 men, 28 women; mean age, 48.1 years, range, 21–78 years) who completed a 1-year follow-up were enrolled. The mean follow-up period was 17.1 months (range, 12–52 months) (Table 1). The causes of the fractures were traffic accidents (n = 34), slips (n = 11), falls (n = 6), sports injuries (n = 10), direct injuries (n = 4), and others (n = 10). Four of these were open fractures (Gustilo-Anderson classification I: 2, II: 1, IIIa: 1) [14]. Based on the Arbeitsgemeinschaft für Osteosynthesefragen/Orthopedic Trauma Association (AO/OTA) classification, there were 30, 38, and 7 cases of 12A, 12B, and 12C fractures, respectively [15]. The study design and data collection were approved by the Institutional Review Board of the Human Experiment and Ethics Committee of our hospital (IRB no. 2022-09-018-001). Informed consent was obtained from the patients and/or their families after they were informed that the data from their cases would be submitted for publication.

### 2.2. PreOperative Evaluation

A proper preoperative evaluation was conducted to determine the most appropriate surgical approach and ensure optimal fixation, Plain radiography was performed in all patients to assess the fracture patterns and measure the length of the distal fragment. Additionally, three-dimensional computed tomography was used to confirm the absence of intra-articular extension and assist in preoperative planning. The length of the distal fragment was measured as the shortest distance from the upper margin of the coronoid fossa to the most distal fracture line on anteroposterior radiographs, as described by Sohn and Shin [16]. To evaluate preoperative nerve function, clinical examination of the radial nerve—including assessment of wrist and finger extension as well as dorsal hand sensation—was performed in the emergency department prior to the application of a long arm splint.

To compare the outcomes of anterior and posterior plating under similar conditions, only patients with distal fragments measuring at least 3 cm from the coronoid fossa were included, based on a previous biomechanical study [13] showing that a minimum of 3 cm is required for secure anterior fixation using four locking screws. Additional inclusion criteria consisted of de novo fractures due to trauma, age of 18 years or older, and normal elbow and shoulder function prior to injury. Exclusion criteria included a history of DHSF, pathological fractures, active malignancy, infections, intra-articular distal humerus fractures, a follow-up period of less than one year, another ipsilateral arm fracture affecting rehabilitation, pre-existing degenerative elbow or shoulder disease, altered consciousness preventing radial nerve testing, and patients under 18 years to exclude bias related to ongoing bone growth.

### 2.3. Surgical Methods

Surgical procedures were performed within two to three days of the injury under general anesthesia. Patients were randomly assigned to undergo either anterior or posterior plating to ensure that both groups met the same preoperative conditions.

For anterior plating, the anterolateral approach was used [17]. The patient was placed in the supine position and an incision was made along the lateral aspect of the biceps brachii. The brachialis muscle was split longitudinally at its mid-portion, and care was taken to dissect between the musculocutaneous and radial nerve territories to prevent denervation. After exposing the fracture site, reduction was achieved using pointed reduction forceps, and a 2.7 mm miniplate was applied as a reduction plate to maintain alignment during definitive fixation [18]. Reduction was confirmed using a C-arm fluoroscope (OEC 9900 Elite, GE Healthcare, Chicago, IL, USA), after which a PHILOS plate (DePuy Synthes, Synthes GmbH, Oberdorf, Switzerland) was applied [16]. The plate was positioned anteriorly on the humerus with its proximal end aligned with the coronoid fossa and secured with at least four screws in the distal main fragment (Figure 1) [19].

For posterior plating, the triceps-sparing approach described by Gerwin et al. [20], was utilized. The patient was placed in either the prone or lateral position with the shoulder abducted and the elbow flexed to facilitate optimal exposure. A midline posterior incision of 10 to 15 cm was made along the humeral shaft, and the triceps brachii was retracted laterally to expose the fracture. Following open reduction, an extra-articular distal humeral plate (EADHP) (DePuy Synthes, Synthes GmbH, Oberdorf, Switzerland) was used (Figure 2).

The plate was positioned along the posterior humeral shaft with its distal end curved along the lateral epicondyle. If necessary, plate bending was performed to optimize the fit. At least four screws were inserted into both the proximal and distal fragments, and lag screws were added in cases with wedge fragments [21].

In both the anterolateral and posterior approaches, plate positioning and fracture reduction were confirmed intraoperatively using fluoroscopy. A suction drain was inserted before performing layered closure of the muscles and skin [22].

### 2.4. Postoperative Rehabilitation

A long arm splint was applied for one week postoperatively. After the initial immobilization period, passive ROM exercises for the elbow and shoulder were initiated based on the patient’s pain tolerance. External and internal rotation exercises were performed two weeks after surgery to facilitate functional recovery.

### 2.5. Medical Records Review

Patients diagnosed with DHSF were classified into the anterior and posterior plating groups. To ensure comparability between the groups, preoperative demographic and clinical factors—including age, sex, body mass index, smoking history, mechanism of injury, injury severity score (ISS), and American Society of Anesthesiologists (ASA) classification—were analyzed to identify potential preoperative statistical biases [23]. Fracture classifications were assigned according to the AO/OTA classification system [15].

The operative time and intraoperative blood loss were assessed using both surgical approaches [24]. Postoperative radiographic evaluations were conducted immediately after surgery, at two weeks, six weeks, three months, and subsequently at three-month intervals. Bone union was defined as the presence of callus formation on at least three of the four cortical bone surfaces accompanied by the absence of pain at the fracture site [25]. The time required to achieve the bone union was also recorded. Non-union was defined as the failure to achieve complete bone healing on radiographs obtained at least six months postoperatively, the absence of progressive bone formation over a three-month follow-up period, or persistent pain at the fracture site [26]. Shortening or angular deformities were classified as varus or valgus angulation of 15° or greater, or a shortening of 3 cm or more [27].

One-year postoperatively, clinical outcomes were assessed using the Mayo Elbow Performance Score (MEPS), which categorizes results as excellent (90–100), good (75–89), fair (60–74), or poor (<60). Outcomes rated as good or excellent were considered satisfactory [10,17].

Complications were retrospectively reviewed and classified as postoperative radial nerve palsy, non-union, infection, implant failure, or the need for revision surgery [22]. Infections were further categorized into superficial infections, which were managed with oral antibiotics, and deep infections, which required surgical intervention including irrigation and debridement. Implant failures and cases requiring revision surgery have also been reported.

### 2.6. Statistical Analysis

Statistical analyses were performed according to the standardized guidelines described by Strage et al. [28]. Descriptive statistics were used to summarize the patient demographics and surgical outcomes. The Shapiro–Wilk test was used to assess the normality of continuous variables. For comparisons between the anterior and posterior plating groups, independent *t*-tests were conducted for normally distributed continuous variables, whereas the Mann–Whitney U-test was used for non-normally distributed data. Pearson’s Chi-square test or Fisher’s exact test was used to evaluate categorical variables, including complication rates, bone union rates, and patient-reported outcomes. The Kruskal–Wallis test was applied to compare non-parametric data between multiple subgroups, if applicable. To examine the relationship between the surgical approach and postoperative complications, Chi-square tests of independence were conducted using 2 × 2 contingency tables. The significance level was set at *p* < 0.05, with results considered statistically significant. IBM SPSS software (version 23.0; IBM Corp., Armonk, NY, USA) was used for statistical analysis.

## 3. Results

Among the 75 patients included in the study, the anterolateral approach was used in 50 patients, while the posterior triceps-sparing approach was used in 25. There were no statistically significant differences between the two groups in terms of preoperative demographic characteristics (*p* > 0.05; Table 1).

The mean length of the distal fragment was 4.31 cm (range: 3.00–9.32 cm, *p* = 0.19). The mean operative time was significantly shorter in the anterolateral group (116 ± 29.4 min) compared to the posterior group (143 ± 31.4 min, *p* = 0.03). Intraoperative blood loss was 253 ± 15.4 mL (range: 198–302 mL) in the anterolateral group and 292 ± 27.9 mL (range: 254–405 mL) in the posterior group, showing no statistically significant difference (*p* = 0.36; Table 2).

Bone union was achieved in 74 out of 75 cases, with an average time to union of 17.5 ± 7.8 weeks (range: 10–32 weeks). One case of non-union occurred in the anterolateral plate group, requiring autologous iliac crest bone grafting and plate augmentation, which resulted in successful bone union. There were no cases of non-union in the posterior plating group. Two cases of iatrogenic radial nerve palsy were observed in the posterior plating group, both of which gradually recovered within three to six months with conservative treatment.

At the one-year follow-up, the mean MEPS was 87.2 ± 9.1 (range: 55–100) in the anterolateral plating group and 81.8 ± 8.4 (range: 55–95) in the posterior plating group, with no statistically significant difference (*p* = 0.71). The mean elbow ROM was 126° ± 5.2° (range: 112–140°) in the anterolateral group and 118° ± 3.1° (range: 90–135°) in the posterior group, also showing no significant difference (*p* = 0.36).

Implant removal was performed in three cases in the posterior plating group. One patient underwent plate removal due to ROM limitation, while two experienced discomfort caused by a bony prominence at the plate site. No implant removal was required in the anterolateral group (*p* = 0.034). The postoperative complications were minimal. Two cases of superficial infection occurred in one patient in each group. Both were successfully treated with oral antibiotics, and no additional surgical interventions were required. There were no cases of deep infection, implant failure, or malalignment (*p* = 0.21). At the final follow-up, none of the patients exhibited severe extension lag or elbow stiffness, although the patients in the posterior plating group reported slightly greater subjective stiffness and mild residual discomfort.

## 4. Discussion

Distal third humeral shaft fractures present unique challenges in surgical management owing to their proximity to the elbow joint, limited bone stock for fixation, and the risk of radial nerve injury [22,24,27]. Although various surgical approaches and fixation methods have been proposed, controversy remains regarding the optimal treatment strategy [29,30]. This study compared the clinical and radiographic outcomes of anterior and posterior plating, while ensuring similar preoperative conditions between the groups.

A minimum distal fragment length of 3 cm was the critical inclusion criterion. This threshold was selected based on biomechanical and anatomical considerations. Finite element analysis has demonstrated that anterior plating with a modified PHILOS plate provides sufficient biomechanical stability when at least four locking screws can be placed in a distal fragment measuring 30 mm or longer [13]. Cadaveric studies further support that anterior plating is feasible when adequate screw purchase can be achieved above the olecranon fossa [31,32]. However, when the distal fragment is shorter than 3 cm, stable fixation using anterior plating becomes challenging, and posterior plating may be preferable [16]. By ensuring that all patients in this study had a distal fragment of at least 3 cm, the potential fixation-related bias was minimized, allowing for a fair comparison between the two approaches.

Among the analyzed variables, intraoperative time was the only factor that showed a statistically significant difference between the two surgical approaches (*p* = 0.03), with the anterolateral approach requiring a shorter operative duration. This discrepancy can be attributed to several factors. First, the anterolateral approach, performed with the patient in the supine position, provides direct access to the anterior surface of the humerus, allowing for more efficient reduction and plating. Additionally, to prevent iatrogenic nerve injury, this approach avoids the need for radial nerve dissection, which is often required in posterior plating [17]. Contrastingly, the posterior triceps-sparing approach necessitates careful handling of the radial nerve and more extensive soft tissue dissection and retraction to achieve proper plate placement in the lateral decubitus or prone position, which may further contribute to increased operative time [24]. These findings are consistent with those of previous studies that reported similar trends [33,34].

Despite the difference in operative time, other surgical and clinical outcomes, including intraoperative blood loss, bone union rates, functional scores, and complication rates, were not significantly different between the two approaches (*p* > 0.05). This suggests that while the anterolateral approach may offer advantages in terms of efficiency, both techniques provide comparable long-term results when applied under appropriate conditions [8,9]. Furthermore, the lack of significant differences in the complication rates, including radial nerve palsy and implant-related issues, reinforces the safety of both approaches.

One of the most significant concerns associated with the posterior approach is the risk of iatrogenic radial nerve palsy [22,35,36]. In this study, two cases of transient radial nerve palsy occurred in the posterior plating group, whereas no such cases were observed in the anterior group. This is consistent with findings from systematic reviews that reported significantly higher incidences of radial nerve palsy with posterior plating, which was attributed to direct nerve manipulation and retraction during surgery [37]. Although both patients in this study recovered fully within three to six months, the potential for transient nerve dysfunction remains a key consideration when choosing a posterior approach.

Implant-related complications are another notable difference between the two approaches. Three patients in the posterior plate group required implant removal, including one due to limited ROM and two due to symptomatic plate prominence. Implant removal was not performed in the anterior group. These findings suggest that posterior plates, particularly those positioned along the lateral column of the humerus, may be more prone to irritation and discomfort, necessitating their removal in some cases. This aligns with previous studies that reported higher hardware-related complications following posterior plating [11]. Functional recovery, assessed using the MEPS and ROM, showed satisfactory outcomes in both groups. However, the anterior plating group exhibited slightly better MEPS and ROM, with a mean MEPS of 87.2° compared with 81.8° in the posterior plating group and a mean elbow ROM of 126° versus 118°, respectively. Although the differences were not statistically significant, the trend suggests that anterior plating may offer better preservation of elbow function. This is likely due to the preservation of the posterior soft tissues, including the triceps, leading to improved postoperative rehabilitation [17]. Given these findings, anterior plating appears to be the preferred option when the distal fragment length allows for secure fixation, offering advantages such as a shorter operative time, reduced blood loss, and a lower risk of radial nerve injury [33,34,37]. However, posterior plating remains a viable alternative in cases where anterior fixation is not feasible due to limited distal bone stock [13].

Despite the strengths of this study, it has several limitations. Although efforts were made to ensure comparability between groups, the retrospective nature of the study introduced a potential selection bias. A randomized controlled trial would provide stronger evidence regarding the optimal surgical approach. Additionally, the follow-up period primarily assessed early union and functional recovery, leaving out long-term complications such as post-traumatic arthritis or late implant failure. Finally, although this study controlled for the distal fragment length, additional biomechanical research is needed to determine the precise threshold at which anterior plating becomes unreliable, particularly in patients with osteoporosis.

## 5. Conclusions

Both the anterolateral and posterior approaches provide effective surgical options for distal humeral shaft fractures, achieving high bone union rates and satisfactory functional outcomes. The anterolateral approach significantly reduces the operative time, likely due to supine positioning, direct fracture access, and the avoidance of radial nerve dissection. However, both techniques demonstrated comparable results in terms of blood loss, bone union, functional recovery, and complication rates. The choice of approach should be based on fracture characteristics, particularly the distal fragment length. Anterior plating is recommended when the distal fragment is at least 3 cm long to ensure stable fixation with minimal surgical trauma. In cases where anterior fixation is not feasible, posterior plating remains a viable alternative. Further studies should evaluate the long-term functional outcomes and biomechanical stability to refine surgical decision-making.

## Figures and Tables

**Figure 1 jcm-14-02890-f001:**
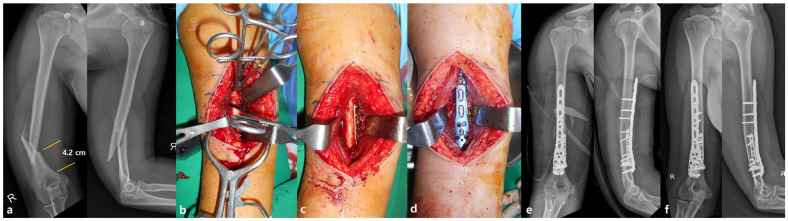
(**a**) Preoperative AP and lateral radiographs showing an AO/OTA 12-B2 type distal humerus fracture with a wedged fragment; the length of the distal fragment was 4.2 cm from the coronoid fossa. (**b**) Intraoperative photograph showing fixation using pointed reduction forceps. (**c**) A 2.7 mm miniplate was applied as a reduction plate and retained as part of the final fixation construct. (**d**,**e**) Anterior plating using a PHILOS plate via the anterolateral approach. (**f**) Postoperative radiograph demonstrating bone union.

**Figure 2 jcm-14-02890-f002:**
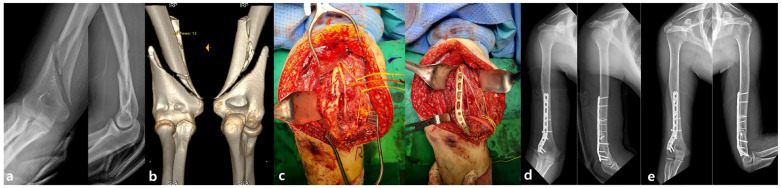
(**a**,**b**) Preoperative AP and lateral radiographs and 3D CT images showing an AO/OTA 12-B2 type distal humerus fracture with a medial wedged fragment. (**c**,**d**) Intraoperative views showing fixation with a lag screw and extra-articular distal humeral plate (EADHP) using the posterior triceps-sparing approach. (**e**) Postoperative radiograph demonstrating fracture union.

**Table 1 jcm-14-02890-t001:** Patient demographics.

Characteristics		Anterolateral Approach (n = 50)	Posterior Triceps-Sparing Approach (n = 25)	*p*-Value
Sex	Male	31	16	1.00
	Female	19	9	
Age	Mean ± SD	46.23 ± 7.81	51.18 ± 10.09	0.45
Medical history	No	37	15	0.33
	Yes	13	10	
Smoking	No	35	21	0.30
	Yes	15	4	
BMI	Mean ± SD	25.23 ± 8.27	27.58 ± 11.36	0.12
ISS	Mean ± SD	6.57 ± 4.28	7.52 ± 5.31	0.88
Injury mechanism	Traffic accident	24	10	0.93
	Slip down	6	5	
	Fall down	4	2	
	Sports injury	6	4	
	Direct injury	3	1	
	Others	7	3	
Open fracture	No	47	24	1.00
	Yes	3	1	
Combined injuries	No	31	17	0.80
	Yes	19	8	
ASA classification	1	17	9	0.96
	2	28	14	
	3	5	2	
AO/OTA classification	12-A	22	8	0.52
	12-B	23	15	
	12-C	5	2	
Distal fragment length	Mean (cm) ± SD	4.87 ± 3.41	4.01 ± 2.78	0.09

**Table 2 jcm-14-02890-t002:** Surgical and clinical outcomes.

Characteristics		Anterolateral Approach (n = 50)	Posterior Triceps-Sparing Approach (n = 25)	*p*-Value
Intraoperative time	Minutes	115.77 ± 29.41	143.15 ± 31.41	**0.03**
Intraoperative blood loss	mL	253.43 ± 15.38	291.72 ± 27.91	0.36
Follow-up	Months	17.11 ± 9.91	15.39 ± 13.12	0.63
Bone union	No	1	0	1.00
	Yes	49	25	
Time to bone union	Weeks	16.99 ± 7.81	19.31 ± 5.37	0.07
Clinical results (Mayo Elbow Performance Score)	Excellent (>90)	25	13	0.71
	Good [75–89]	13	8	
	Fair [60–74]	11	3	
	Poor (<60)	1	1	
Elbow range of motion	Mean ± SD	126.34 ± 5.19	118.74 ± 3.11	0.36
Complications	Non-union	1	0	0.21
	Postoperative Radial nerve injury	0	2	
	Superficial infection	1	1	
	Plate irritation	0	2	

The bold font was used to highlight the only *p*-value below 0.05, indicating statistical significance.

## Data Availability

The datasets generated and/or analyzed during the current study are not publicly available because of restricted access to our hospital database but are available from the corresponding author upon reasonable request.

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
