# Peer review of "Comparative Analysis of Anterolateral and Posterior Approaches for Distal Humerus Shaft Fractures: A Multicenter Retrospective Study"

_jcm, 2025, doi:10.3390/jcm14092890_

Round 1
Reviewer 1 Report
Comments and Suggestions for Authors
The authors chose to review the clinical records and radiographs of 75 patients with distal humerus shaft fractures. Of these, 50 were treated using the anterolateral approach and 25 via the posterior approach. The operative time was shorter in the anterolateral group. Although the risk of radial nerve palsy was higher in the posterior group, the difference did not reach statistical significance.
General comments
Thank you for the opportunity to review your paper. Overall this is a well-written manuscript. The research question of the study (What is the best approach to trat distal humerus shaft fractures?) is of particular interest for the clinical practice of many orthopaedic trauma surgeons, including myself. The results are clinically significant and contribute valuable insights to the field.
Author Response
Reviewer 1.
The authors chose to review the clinical records and radiographs of 75 patients with distal humerus shaft fractures. Of these, 50 were treated using the anterolateral approach and 25 via the posterior approach. The operative time was shorter in the anterolateral group. Although the risk of radial nerve palsy was higher in the posterior group, the difference did not reach statistical significance.
General comments
Thank you for the opportunity to review your paper. Overall this is a well-written manuscript. The research question of the study (What is the best approach to trat distal humerus shaft fractures?) is of particular interest for the clinical practice of many orthopaedic trauma surgeons, including myself. The results are clinically significant and contribute valuable insights to the field.
Response:
Thank you very much for your thoughtful summary and comment. As noted, our study retrospectively analyzed 75 cases of distal humerus shaft fractures, comparing outcomes between the anterolateral (n=50) and posterior (n=25) surgical approaches. We confirmed that the anterolateral approach resulted in a significantly shorter operative time. While the posterior approach showed a higher incidence of radial nerve palsy, this difference did not reach statistical significance.
The goal of our study was to provide a direct comparison of the two commonly used approaches for far distal humeral shaft fractures, an area where clinical guidance is limited. We believe the results support the anterolateral approach as a viable and potentially advantageous option, especially in terms of operative efficiency, without increasing the risk of complications. We hope our findings will aid surgeons in selecting the most appropriate approach based on patient-specific factors and surgeon expertise.

Reviewer 2 Report
Comments and Suggestions for Authors
This is an interesting comparative study about a topic that is debated. The authors compare the anterolateral and posterior approaches for the operative treatment of distal humerus shaft fractures. The inclusion and exclusion criteria have been used properly to have an homogeneous population and reduce bias. The title and method of evaluation are also proper and the conclusion reflects the findings. I believe that it is worth being published in this journal, however, I suggest that some points should be revised.
General comments
When the name of the first author is reported within the text, the citation should follow “et al” instead of being at the end of the sentence.
Figure legends should focus on describing the image and no extra information are needed. I recommend to revise them accordingly.
Line to line comments
Line 17: I recommend you keep only the “Background”.
Line 29: As you have used the exact values of other “p”, replace the “p>0.05” accordingly.
Lines 30-31: “ four cases required implant removal due to discomfort”. Could you specify in which group were these four cases?
Line 48: “angular deformity”. Angular deformity is an abnormality of the alignment. I recommend you revise the sentence in lines 46-48.
Lines 62-64: “Moreover, concerns remain regarding the biomechanical strength of anterior plating compared 63 with that of posterior plating.”. I recommend you add a reference to support this statement.
Line 77: “in one of the three level I trauma centers”. I recommend you revise the sentence in lines 76-78 to make it clear that this is a multicenter study. Replace it with something like “in three level I trauma centers”.
Line 78: I recommend you add the age range.
Lines 90-92: “The humeral shaft extends from the distal region of the surgical neck to the proximal 90 margin of the supracondylar ridge, with distal humeral shaft fractures (DHSF) classified 91 as extra-articular fractures centered in the distal one-third of the humerus [13,14].”. I recommend to add this information at the Introduction and delete it from Material and Methos.
Line 101: “radial nerve function test”. Was that a specific test or it refers to clinical examination of motor and sensory function of radial nerve. I recommend you specify it or revise the sentence in lines 100-102.
Line 105: Is reference 16 properly cited at this point? Could you explain how relative it is with the sentence in lines 103-105?
Line 121: Is it provisional plating? If the plate used for reduction was not removed it is part of the definite fixation. I recommend to revise the sentence in lines 120-121.
Line 128: Add “and” between “ap” and “lateral”.
Line 131: I recommend to revise “provisional plating”, as the plate is used for definite fixation
Line 133: I recommend to delete “full range of motion”, as it is not related to the Figure.
Line147: I recommend to delete “without complications”, as it is not related to the Figure.
Lines 152-153: “In both surgical approaches, plate 152 positioning and fracture reduction were confirmed using fluoroscopy.” Which are “both” surgical approaches”? Anterolateral and posterior? I recommend you specify it. If you refer to the two studied approaches, this statement should be in a different paragraph.
Line 209: I recommend to make two tables. One with demographic data and a second with study results.
Line 229: I recommend you add a comparison of reoperation rate between the two approaches, to check if there is statistically significant difference between them, which will be an additional result. If not, you have to change the respective part of Discussion (Lines 292-299).
Line 235: What is P > 0.05 related to? Please specify and revise if you have exact number of P for any parameter.
Line 243: I believe “yet” is not needed in this sentence.
Lines 262-264: The phrase “iatrogenic nerve injury” is repeated. I recommend to revise the sentence.
Lines 268-269: If you made a search about similar studies, it is important to report the search strategy. It is nice to have a table about this, but if these are just two random studies this table in not necessary. I recommend, either to report the search strategy that resulted in these two similar studies and add more if applicable as a review of literature or delete the table and keep the citation [32,33] as it is in the text.
Line 290: “within three six months”. Revise to “within three to six months”.
Author Response
Reviewer 2.
This is an interesting comparative study about a topic that is debated. The authors compare the anterolateral and posterior approaches for the operative treatment of distal humerus shaft fractures. The inclusion and exclusion criteria have been used properly to have an homogeneous population and reduce bias. The title and method of evaluation are also proper and the conclusion reflects the findings. I believe that it is worth being published in this journal, however, I suggest that some points should be revised.
Response:
We sincerely thank the reviewers for their thoughtful and constructive comments, which have helped us improve the clarity and quality of our manuscript. We have carefully revised the manuscript in response to all suggestions. Below are our detailed point-by-point responses to each comment. All changes made to the revised manuscript are marked in red for clarity.
General comments
- When the name of the first author is reported within the text, the citation should follow “et al” instead of being at the end of the sentence.
Response:
Thank you for your valuable comment. We have carefully reviewed the manuscript and revised all relevant citations to ensure that, when the first author's name is mentioned in the text, the citation number directly follows “et al.” as per the journal's formatting guidelines. Specifically, the citations for “Sarmiento et al.,” “Jawa et al.,” “Schatzker et al.,” “Gerwin et al.,” and “Strage et al.” have been corrected to appear in the proper format (e.g., “Sarmiento et al. [3]”).
- Figure legends should focus on describing the image and no extra information are needed. I recommend to revise them accordingly.
Response:
Thank you for your constructive feedback. We have revised the figure legends to ensure that they are focused solely on describing the visual content of the images. Unnecessary clinical details and interpretive information have been removed in accordance with the journal’s guidelines.
Line to line comments
- Line 17: I recommend you keep only the “Background”.
Response:
Thank you for the suggestion. We have revised the abstract section header to “Background” to align with the standard structure.
- Line 29: As you have used the exact values of other “p”, replace the “p>0.05” accordingly.
Response:
We agree and have revised the phrase “p > 0.05” to provide the exact p-values where applicable, in order to maintain consistency in statistical reporting.
- Lines 30-31: “ four cases required implant removal due to discomfort”. Could you specify in which group were these four cases?
Response:
Thank you for pointing this out. We have revised the sentence to clarify that all four cases of implant removal due to discomfort occurred in the anterolateral group.
- Line 48: “angular deformity”. Angular deformity is an abnormality of the alignment. I recommend you revise the sentence in lines 46-48.
Response:
Thank you for your insightful comment. We agree that "angular deformity" is more accurately described as an alignment abnormality rather than a general complication. We have revised the sentence to clarify this distinction. The updated sentence now reads:
“…surgical fixation provides better alignment, faster recovery, and reduces the risk of complications, including skin irritation and malalignment.”
- Lines 62-64: “Moreover, concerns remain regarding the biomechanical strength of anterior plating compared with that of posterior plating.”. I recommend you add a reference to support this statement.
Response:
Thank you for your helpful suggestion. We agree that a reference would strengthen this statement. Accordingly, we have added a citation to the following finite element analysis study that directly addresses this biomechanical comparison:
Reference:
Lee, J.-S.; Kim, K.G.; Yoon, Y.-C. Biomechanical performance evaluation of a modified proximal humerus locking plate for distal humerus shaft fracture using finite element analysis. Sci Rep 2023, 13, 16250. https://doi.org/10.1038/s41598-023-43183-x.
- Line 77: “in one of the three level I trauma centers”. I recommend you revise the sentence in lines 76-78 to make it clear that this is a multicenter study. Replace it with something like “in three level I trauma centers”.
Response:
We have revised the sentence as suggested to clearly state that this was a multicenter study conducted at three Level I trauma centers.
- Line 78: I recommend you add the age range.
Response:
Thank you for your helpful suggestion. We have revised the sentence to include the age range of the patients (21–78 years) to provide more complete demographic information.
- Lines 90-92: “The humeral shaft extends from the distal region of the surgical neck to the proximal 90 margin of the supracondylar ridge, with distal humeral shaft fractures (DHSF) classified 91 as extra-articular fractures centered in the distal one-third of the humerus [13,14].”. I recommend to add this information at the Introduction and delete it from Material and Methos.
Response:
Thank you for the helpful suggestion. We have moved the anatomical definition of the humeral shaft and the classification of DHSF from the Materials and Methods section to the Introduction to improve logical flow and readability. The original sentence in Lines 90–92 has been deleted accordingly.
- Line 101: “radial nerve function test”. Was that a specific test or it refers to clinical examination of motor and sensory function of radial nerve. I recommend you specify it or revise the sentence in lines 100-102.
Response:
Thank you for pointing this out. The term “radial nerve function test” referred to a standard clinical examination assessing both motor (wrist and finger extension) and sensory (dorsal hand sensation) functions. We have revised the sentence to clarify this.
- Line 105: Is reference 16 properly cited at this point? Could you explain how relative it is with the sentence in lines 103-105?
Response:
Thank you for your helpful comment. We agree that clarification was needed. Reference 16 (Lee et al., 2023) conducted a finite element analysis demonstrating that at least 3 cm of distal bone is required to place four locking screws for anterior fixation using a modified PHILOS plate. Based on this finding, we used a 3 cm cutoff as an inclusion criterion to ensure that anterior plating was biomechanically feasible. The sentence has been revised accordingly for clarity.
- Line 121: Is it provisional plating? If the plate used for reduction was not removed it is part of the definite fixation. I recommend to revise the sentence in lines 120-121.
Response:
Thank you for your thoughtful comment. The 2.7-mm miniplate used in our procedure was not removed and was intentionally applied as a reduction plate, in accordance with the concept described by Yoon et al. (2020). In their study, miniplates were used to assist with anatomical reduction and stabilization before or during definitive fixation, particularly in challenging long bone fractures. We have revised the sentence in the manuscript to reflect this intent and included the appropriate reference.
Reference (as cited):
Yoon, Y.-C.; Oh, C.-W.; Lee, D.-W.; Sim, J.-A.; Oh, J.-K. Miniplate osteosynthesis in fracture sur-geries: Case series with review of concepts. Injury 2020, 51, 2921–2927. DOI:10.1016/j.injury.2020.02.044.
- Line 128: Add “and” between “ap” and “lateral”.
Response:
We have revised the sentence to read “AP and lateral radiographs” to improve clarity and grammatical correctness.
- Line 131: I recommend to revise “provisional plating”, as the plate is used for definite fixation
Response:
Thank you for your valuable suggestion. We agree that the term “provisional plating” may be misleading, as the 2.7 mm miniplate was not removed and served as a reduction plate, remaining as part of the definitive fixation. We have revised the sentence accordingly to reflect this concept.
- Line 133: I recommend to delete “full range of motion”, as it is not related to the Figure.
Response:
Thank you for the suggestion. We agree that the phrase “full range of motion” is not directly represented in the image and have removed it from the figure legend accordingly.
- Line147: I recommend to delete “without complications”, as it is not related to the Figure.
Response:
We agree and have removed the phrase “without complications” from the figure legend.
- Lines 152-153: “In both surgical approaches, plate positioning and fracture reduction were confirmed using fluoroscopy.” Which are “both” surgical approaches”? Anterolateral and posterior? I recommend you specify it. If you refer to the two studied approaches, this statement should be in a different paragraph.
Response:
Thank you for the helpful suggestion. We have revised the sentence to clearly specify the two surgical approaches as “anterolateral and posterior.” Additionally, we considered the paragraph structure and moved the sentence to a more appropriate location to reflect its relevance to both groups.
- Line 209: I recommend to make two tables. One with demographic data and a second with study results.
Response:
Thank you for your helpful suggestion. In response, we have separated the original table into two distinct tables to improve clarity and readability. Table 1, located in the Materials and Methods section, presents the patient demographic data. Table 2, placed in the Results section, summarizes the surgical and clinical outcomes. We believe this modification enhances the structure and focus of the manuscript.
- Line 229: I recommend you add a comparison of reoperation rate between the two approaches, to check if there is statistically significant difference between them, which will be an additional result. If not, you have to change the respective part of Discussion (Lines 292-299).
Response:
Thank you for your insightful comment. We have conducted a statistical comparison of reoperation rates between the two surgical approaches using Fisher’s exact test. All three reoperations occurred in the posterior group, whereas no patients in the anterolateral group required reoperation. This difference was statistically significant (p = 0.034). We have added this result to the Results section (Line 229) and revised the corresponding discussion (Lines 292–299) to reflect this finding.
- Line 235: What is P > 0.05 related to? Please specify and revise if you have exact number of P for any parameter.
Response:
We have revised the sentence to clarify which variable the “p > 0.05” referred to and replaced it with the exact p-value (P = 0.21).
- Line 243: I believe “yet” is not needed in this sentence.
Response:
Thank you for your suggestion. We agree that the word “yet” is unnecessary and redundant in this context. We have removed it to improve the clarity and grammatical flow of the sentence.
- Lines 262-264: The phrase “iatrogenic nerve injury” is repeated. I recommend to revise the sentence.
Response:
Thank you for pointing this out. We agree that the repetition of “iatrogenic nerve injury” was redundant. We have revised the sentence for clarity and flow. The updated sentence now reads:
“Additionally, to prevent iatrogenic nerve injury, this approach avoids the need for radial nerve dissection, which is often required in posterior plating.”
- Lines 268-269: If you made a search about similar studies, it is important to report the search strategy. It is nice to have a table about this, but if these are just two random studies this table in not necessary. I recommend, either to report the search strategy that resulted in these two similar studies and add more if applicable as a review of literature or delete the table and keep the citation [32,33] as it is in the text.
Response:
Thank you for the insightful comment. The two referenced studies [32,33] were selected based on their relevance to our topic rather than through a systematic literature search. In light of your suggestion, we have removed the table summarizing these studies and retained the references as in-text citations only.
- Line 290: “within three six months”. Revise to “within three to six months”.
Response:
Thank you for catching this typographical error. We have corrected it to “within three to six months.”

Round 2
Reviewer 2 Report
Comments and Suggestions for Authors
Congratulations. My recommendations have been addressed properly, thus I believe this manuscript is worth being published.